# The Optimized Multi-Scale Permutation Entropy and Its Application in Compound Fault Diagnosis of Rotating Machinery

**DOI:** 10.3390/e21020170

**Published:** 2019-02-12

**Authors:** Xianzhi Wang, Shubin Si, Yu Wei, Yongbo Li

**Affiliations:** 1School of Mechanical Engineering, Northwestern Polytechnical University, Xi’an 710072, China; 2Department of Astronautical Science and Mechanics, Harbin Institute of Technology (HIT), Harbin 150001, China; 3School of Aeronautics, Northwestern Polytechnical University, Xi’an 710072, China

**Keywords:** rotating machinery, parameter selection, multi-scale permutation entropy, mutual information, improved Cao method

## Abstract

Multi-scale permutation entropy (MPE) is a statistic indicator to detect nonlinear dynamic changes in time series, which has merits of high calculation efficiency, good robust ability, and independence from prior knowledge, etc. However, the performance of MPE is dependent on the parameter selection of embedding dimension and time delay. To complete the automatic parameter selection of MPE, a novel parameter optimization strategy of MPE is proposed, namely optimized multi-scale permutation entropy (OMPE). In the OMPE method, an improved Cao method is proposed to adaptively select the embedding dimension. Meanwhile, the time delay is determined based on mutual information. To verify the effectiveness of OMPE method, a simulated signal and two experimental signals are used for validation. Results demonstrate that the proposed OMPE method has a better feature extraction ability comparing with existing MPE methods.

## 1. Introduction

Permutation entropy [1,2], as a statistic indicator to detect nonlinear dynamic changes, has been widely used in the fault feature extraction of rotating machinery [3,4,5,6]. For comprehensive analysis, multi-scale permutation entropy (MPE) is proposed to enhance the ability of character description of permutation entropy (PE) [7,8,9]. MPE has obvious merits in describing the complexity in time series, such as high calculation efficiency, good robust ability, and independence from prior knowledge, etc. Recently, more attention has been paid to the application of MPE in fault diagnosis of rotating machinery [10,11,12,13]. Zhao et al. [14] applied MPE and the hidden Markov model to identify different fault types of the rolling bearings. Wu et al. [15,16] used MPE to extract the fault features and then applied a support vector machine to identify the bearing fault types. Tiwari et al. [17] combined MPE with an adaptive neuro fuzzy classifier to recognize incipient bearing faults. Zheng et al. [18] utilized the MPE and support vector machines to recognize various bearing faults. Yao et al. [19] employed the MPE to describe the fault characteristics and then used the extreme learning machine for bearing pattern identifications.

These works have successfully applied MPE in fault diagnosis of rotating machinery. However, the performance of MPE is dependent on the parameters setting, including embedding dimension m and time delay τ. If the embedding dimension m and time delay τ are too small, the reconstruction vector will contain few states, which cannot reflect the real dynamic characteristics of the signal [20,21]. On the other hand, if embedding dimension m and time delay τ are too big, the reconstructed phase space will homogenize the time series, leading to an inappropriate entropy value estimation [22]. Therefore, the parameter selection of embedding dimension m and time delay τ plays an important role in the MPE method [2]. Until now, most researchers have selected MPE parameters according to their experience, thereby, an automatic selection of MPE parameters has become an urgent issue to be solved.

Aiming to automatically select the optimum parameters of MPE, a novel parameter optimization strategy of MPE is proposed in this paper. We call this method optimized multi-scale permutation entropy (OMPE). In the OMPE method, an improved Cao method is proposed to adaptively select embedding dimension m [23]. Meanwhile, the time delay τ is determined based on mutual information (MI) [24]. To verify the effectiveness of the OMPE method, a simulated signal and two experimental signals are used for validation. Results demonstrate that the proposed OMPE method has a better feature extraction ability compared with existing MPE methods.

The organization of the rest of this paper is as follows. Section 2 describes the proposed OMPE method. Section 3 validates the advantages of OMPE using one simulated signal. The effectiveness of the OMPE method is validated using two experimental cases in Section 4. Finally, Section 5 summarizes the conclusions.

## 2. Optimized Multi-Scale Permutation Entropy

Permutation entropy (PE), as a dynamic indicator, is proposed to reflect non-linear behavior of time series by Bandt and Pompe [1]. PE has obvious merits of high calculation efficiency, sensitivity to the dynamic changes, and robustness to noise in estimating the complexity of time series. However, PE confronts challenges in the field of fault feature extraction due to the fault information being embedded in large frequency bands in the vibration signals of rotating machinery. To overcome this shortcoming, multi-scale permutation entropy (MPE) is proposed to enhance the fault feature extraction ability of PE, which can measure the complexity of time series over multi scales. 

Since the order of the amplitude is considered in the phase space that is reconstructed by the embedding dimension m and time delay τ, the determination of the embedding dimension m and time delay τ play an important role in the final performance of MPE. In this study, an optimized multi-scale permutation entropy (OMPE) is proposed to free the parameter selection of MPE. First, the time delay is calculated using mutual information. Based on the obtained time delay τ, we determine the embedding dimension m based on an improved Cao method. Details are described in Section 2.1 and Section 2.2.

### 2.1. Time Delay Calculation Based on Mutual Information

Mutual information (MI) utilizes the information change between the reconstruction space and original time series to select the optimal time delay [24]. Since MI has merits of easy to perform, high calculation efficiency, etc., this paper introduces the MI method to select the optimal time delay of MPE.

For an arbitrary signal $X=\{x_{i},i=1,2,\cdots ,N\}$, it can be reconstructed by time delay τ as follows:(1)$$Y=\{y_{i}\}=\{x_{i},x_{i+\tau }\},\;i=1,2,\cdots ,N-\tau ,$$

According to Ref. [25], the Shanon entropy of X and Y can be calculated as:(2)$$H\left(X\right)=-\sum_{i=1}^{N}p\left(x_{i}\right)\log_{2}\left(p\left(x_{i}\right)\right),$$
(3)$$H\left(Y\right)=-\sum_{i=1}^{n-\tau }p\left(y_{i}\right)\log_{2}\left(p\left(y_{i}\right)\right),$$
where the $p\left(x_{i}\right)$ represents the probability that $X=x_{i}$, and $p\left(y_{i}\right)$ represents the probability that $Y=y_{i}$.

Then, the mutual information between X and Y can be expressed as:(4)$$\begin{array}{lll} I(X,Y) & = & H\left(Y\right)-H\left(Y|X\right)\\ & = & -\sum_{i=1}^{N}\sum_{j=1}^{N-\tau }p\left(x_{i},y_{j}\right)\log_{2}\left[\frac{p\left(x_{i},y_{j}\right)}{p\left(x_{i}\right)p\left(y_{j}\right)}\right] \end{array},$$

The mutual information I(X,Y) measures the information change between the reconstruction space and original time series, which can be utilized to determine the time delay τ. According to [24], the first local minimum point of mutual information I(X,Y) under different time delay τ is the optimum time delay. 

### 2.2. Embedding Dimension Calculation Based on Improved Cao Method

Cao method [23] utilizes the distance variation of the nearest neighbor points in the reconstruction phase space to determine the optimum embedding dimension. It is regarded as an improvement of the false nearest neighbor (FNN) method [26], as it can estimate more appropriate embedding dimension than the traditional FNN method [23]. In the original Cao method, when the average distance of the neighbor points stops changing at some embedding dimension $m_{0}$, the $m_{0}+1$ is the optimal embedding dimension. However, the embedding dimension $m_{0}$ is determined by empirical experience in the original Cao method, which is manual and unsuitable for big data analysis or intelligent fault diagnosis. Therefore, an improved Cao method with adaptive threshold is proposed to automatically determine the optimal embedding dimension. In this paper, the Chebyshev distance is used to measure the performance of MPE with a different embedding dimension m. The embedding dimension m with the largest Chebyshev distance represents that the MPE values have the largest class separation distance. Therefore, the proposed OMPE can extract the most discriminative MPE value as fault features. The whole procedures of the proposed improved Cao method can be divided into two parts: embedding dimension calculation (in Section 2.2.1.) and threshold adjustment (in Section 2.2.2.). 

#### 2.2.1. Embedding Dimension Calculation 

Based on the MI principle, we can select the time delay τ. On the basis of selected time delay τ, the embedding dimension can be determined using the Cao method in the following steps:

Step 1: For an arbitrary signal $X=\{x_{i},i=1,2,\cdots ,N\}$, it can be reconstructed by time delay τ and embedding dimension m as follows:(5)$$Y(m)=y_{i}(m)=\{x_{i},x_{i+\tau },\cdots ,x_{i+(m-1)\tau }\},\;i=1,2,\cdots ,N-(m-1)\tau$$

Note that when m=1, Equation (5) equals Equation (1).

Step 2: Calculate the distance a(i,m) of neighbor points in the phase space.
(6)$$a(i,m)=\frac{\left||y_{i}(m+1)-y_{n(i,m)}(m+1)|\right|}{\left||y_{i}(m)-y_{n(i,m)}(m)|\right|}\;,\;i=1,2,\cdots ,N-m\tau ,$$
where n(i,m)
(1≤n(i,m)≤N−mτ) is an integer such that $y_{n(i,m)}(m)$ is the nearest neighbor of $y_{i}(m)$ in the *m*-dimensional reconstructed phase space. The ||•|| represents the Chebyshev distance as expressed in Equation (7).
(7)$$\left||y_{s}(m)-y_{l}(m)||=\underset{0\leq j\leq m-1}{\max}|x_{s+j\tau }-x_{l+j\tau }\right|$$

Step 3: Calculate the average of all a(i,m) using Equation (8).
(8)$$E(m)=\frac{1}{N-m\tau }\sum_{i=1}^{N-m\tau }a(i,m)$$

Step 4: Use E(m) to calculate E1(m) as expressed in Equation (9).
(9)$$E1(m)=\frac{E(m+1)}{E(m)}$$

In the original Cao method, E1(m) stops changing when m is greater than some value $m_{0}$. Then $m_{0}+1$ is the desired embedding dimension. However, Cao doesn’t describe how to determine $m_{0}$. Hence, we propose a new threshold to determine the embedding dimension in Step 5.

Step 5: Define the increment ΔE1(m) as follows:(10)ΔE1(m)=E1(m+1)−E1(m)
when ΔE1(m+1)≤kΔE1(m), m+1 is the desired embedding dimension. Here, k is the threshold of E1(m). In other words, k determines the embedding dimension. If k is too big, the embedding dimension will be too low, the reconstruction vector will contain too few states which cannot reflect the real dynamic characteristics of the signal; If k is too small, the embedding dimension will be too high, the reconstructed phase space will homogenize the time series resulting inappropriate estimation [22]. Therefore, a novel strategy is proposed to obtain appropriate parameter k in this paper.

#### 2.2.2. Threshold Adjustment using Chebyshev Distance

To obtain an appropriate parameter k, a novel strategy is proposed in this paper. In this method, the Chebyshev distance is used to measure the performance of MPE under different parameter k’s. The parameter k with the largest Chebyshev distance is the optimum threshold for the improved Cao method. 

Suppose that we have obtained the time delay τ according to Section 2.1., the parameter k of proposed improved Cao method contains six calculation steps as following.

Step 1: Set the maximum searching range $k_{m}$ and stride $k_{s}$, and initialize parameter $k=k_{0}$. 

Step 2: Calculate embedding dimension $m_{k}$ according to Section 2.2.1.

Step 3: Calculate the MPE value using the obtained τ and $m_{k}$.

Step 4: Suppose there are c categories training samples, let the $mpe_{a}(m_{k},\tau ,S)$ represent the MPE value of category a using scale S, time delay τ and embedding dimension $m_{k}$. Calculate the Chebyshev distance $D_{ab}$ between each categories according to Equation (11), then it will be c(c−1)/2 distance.
(11)$$D_{ab}=||mpe_{a}-mpe_{b}||=\underset{0\leq j\leq S}{\max}|mpe_{a}(m_{k},\tau ,j)-mpe_{b}(m_{k},\tau ,j)|$$

Step 5: Calculate the average distance $D^{k}$ according to Equation (12).
(12)$$D^{k}=\frac{2}{c(c-1)}\sum \sum D_{ij}$$

Here,$D^{k}$ represents the average distance between different categories, which can be regarded as the class separation distance. The parameter k with the largest $D^{k}$ represents that the MPE values have the largest class separation distance. 

Step 6: Repeat step 1–5 to calculate all the $D^{k}$ under each parameter k. The $k_{opt}$ corresponding to the maximum $D^{k}{}_{\max}$ is the optimum threshold for the improved Cao method. 

The whole procedures of proposed improved Cao method can be seen in Figure 1.

### 2.3. Procedures of OMPE Method

Generally, four steps are required in our proposed OMPE as follows:

Step 1: Apply the MI method to calculate the time delay of MPE as mentioned in Section 2.1. Then the optimal time delay $\tau_{opt}$ can be obtained.

Step 2: Determine the threshold of the proposed improved Cao method which is described in Section 2.2.2. Then the optimal threshold $k_{opt}$ can be obtained.

Step 3: Utilize the obtained threshold $k_{opt}$, calculate the optimal time embedding dimension $m_{opt}$ of MPE by the proposed improved Cao method as mentioned in Section 2.2.1.

Step 4: Calculate the MPE value using the obtained optimum time delay $\tau_{opt}$ and optimum embedding dimension $m_{opt}$.

The flowchart of the OMPE is illustrated in Figure 2.

The proposed OMPE can automatically and adaptively select the optimum time delay $\tau_{opt}$ and embedding dimension $m_{opt}$ of MPE for an arbitrary signal. Since the adaptive threshold uses the Chebyshev distance to choose the embedding dimension m with the largest class separation distance, the proposed OMPE can extract the most discriminative MPE value as fault features, resulting in higher classification accuracy. 

## 3. Simulation Validation

In this section, three bearing signals with different fault types are simulated for validation, including ball fault (BF), inner race fault (IRF) and outer race fault (ORF). The details of the three bearing theoretic models can be referred to in [27]. 

Here, we set the sampling frequency at 20,480 Hz, the data length *N* = 2048 points, the rotating frequency $f_{r}=50\;\mathrm{Hz}$, intrinsic frequency $f_{n}=2548\;\mathrm{Hz}$, outer race fault characteristic $f_{o}=203.9\;\mathrm{Hz}$ with added noise SNR = −18.626 dB, inner race fault characteristic $f_{I}=296.1\;\mathrm{Hz}$ with added noise SNR = −27.91 dB, and ball fault characteristic $f_{B}=262\;\mathrm{Hz}$ with added noise SNR = −36.15 dB. Figure 3 shows the time domains of three simulated bearing signals and their corresponding noised signals. 

The proposed method is applied to analyze these simulated signals as described in Section 2. First, the searching area of parameter k is set from 0.4 to 2.0, and the stride is set to be 0.1. The threshold *k* is used to determine the embedding dimension of MPE. If *k* is set smaller than 0.4, the embedded dimension of MPE will be too large. This will cause that the MPE value cannot be obtained. Otherwise, if *k* is set larger than 2, a smaller embedding dimension of MPE will be generated, resulting in an inappropriate MPE values for complexity estimation. Based on the above reasons, the range of parameter *k* is set from 0.4 to 2. The result of parameter k is plotted in Figure 4. As seen, the maximum value is 0.3265 and the corresponding k=0.5. Therefore, we set k=0.5. Second, the MI and improved Cao method is used to calculate the time delay τ and embedding dimension m. The optimum embedding dimension and optimum time delay is shown in Table 1. Finally, the MPE can be calculated using the optimum embedding dimension and optimum time delay.

For comparison, other parameter selection methods are also used to calculate the MPE value. The first approach uses fixed parameter m=6 and τ=1 (called the FIX method) as described in [18]. The second approach is to use MI to calculate the time delay τ and false nearest neighbor (FNN) to calculate embedding dimension m (called as MI-FNN method) [28]. Table 1 lists the parameter settings of FIX and MI-FNN methods. As seen in Table 1, it can be found that the embedding dimension m of FIX and MI-FNN methods are kept a constant value of 6 and 4, respectively. Only the proposed OMPE method can adaptively select suitable embedding dimension m (6, 9 and 5) for different bearing vibration signals. 

The MPE values of three methods are plotted in Figure 5. As seen in Figure 5, it can be found that the MPE values of three simulated signals can be clearly discriminated in each scale using the proposed method. However, the MPE values can be barely discriminated in each scale using the FIX method or MI-FNN method. This phenomenon can be explained in the following way. First, the OMPE uses different parameter combinations for different kinds of signals, which can better reflect the real dynamic characteristics of the signal. Second, the proposed OMPE method uses Chebyshev distance to choose the embedding dimension m with largest class separation distance, therefore, the OMPE method can discriminate the three bearing fault types significantly. 

## 4. Experimental Validation

In this section, two experiments are designed to prove the effectiveness of the proposed method. In Experiment 1, a compound gear fault experiment is designed. In Experiment 2, a compound bearing and unbalance rotor fault experiment is designed. Meanwhile, the FIX method and MI-FNN method are also used for comparison. For the purpose of a fair comparison, five conditions are the same as follows: (1) 10 scales MPE are used as input features; (2) the k-nearest neighbor (KNN) classifier [29,30] is used as classifier to identify the different fault types; (3) the length of each sample is 2048 points; (4) the KNN is trained by 50% of samples, and the rest samples are utilized to test the performance; (5) each method runs 20 times to reduce the randomness. 

### 4.1. Experiment 1: Compound Gear Fault

In this section, compound gear fault experiments were conducted on rotating machinery called SpectraQuest Machinery Fault Simulator (MFS). The test rig is shown in Figure 6a,b. It consists of a reliance electric motor driving a three-way gearbox with straight cut bevel gears. A magnetic clutch (also called load) is mounted at the rear of the gearbox. In this paper, the load was 5 in-lbs of torque. The different gear faults were simulated by replaced the fault gear (including pitting in the driving tooth, the broken tooth in the driving tooth, the missing tooth in the driving tooth and crack in the follower tooth, shows in Figure 7). An accelerometer was installed on the top of the gearbox to collect the vibration signals. The sampling frequency was 12800 Hz and the rotating speed was 3000 rpm.

In Experiment 1, eight fault types were introduced to the rotating machinery, including Normal condition (NOR), pitting in the driving tooth (PT), broken tooth in the driving tooth (BT), missing tooth in the driving tooth (MT), crack in the follower tooth (CT), pitting in the driving tooth with crack in the follower tooth (PC), broken tooth in the driving tooth with crack in the follower tooth (BC), and missing tooth in the driving tooth with crack in the follower tooth (MC). Each class has 100 samples and there are a total of 800 samples (100 samples × eight fault types). The waveforms under different fault types are shown in Figure 8. The class labels of eight fault types in Experiment 1 are shown in Table 2.

The proposed method is used in this test as described in Section 2. First, the searching area of parameter k is set from 0.4 to 2.0, and the stride is set to 0.1. The result of parameter k is shown in Figure 9. As seen the maximum value is 0.2205 and the corresponding k=0.5. Therefore, we set k=0.5. Second, the MI and improved Cao method is used to calculate the time delay τ and embedding dimension m. The optimum embedding dimension and optimum time delay are shown in Table 2. Third, the MPE value is calculated using the optimum embedding dimension and optimum time delay. Last, the MPE value is used as the input of KNN classifier to identify different fault types. The mean testing accuracy of the proposed method is 99.38%. One of the final classification results using the proposed method is drawn in Figure 10 (testing accuracy 159/160 = 99.38%).

For comparison, the FIX method and MI-FNN method are also tested. In addition, the traversal method is also used for comparison (called the TM method). In the TM method, we test MPE with different combinations of embedding dimension m and time delay τ. The ranges of embedding dimension *m* and τ are set as *m* = 3 to 7 and τ=1 to 10. The result of TM method is shown in Figure 11. The parameters with the highest testing accuracy are used for comparison, that is 99.30% with m=6 and τ=2. The testing accuracies of four methods are shown in Figure 12. 

From Experiment 1, five conclusions can be drawn as follows: First, from Figure 11, we can find the performance of MPE is indeed influenced by embedding dimension m and time delay τ as we mentioned in Section 1. Second, the FIX method performs the worst of the four methods. In [1], Bandt and Pompe recommend *m* = 3 to 7 and τ=1. However the MPE with time delay 2 and 3 performs better than that with time delay 1. This can prove the optimum parameters of MPE are not always in the range of Bandt and Pompe’s recommendation. Third, the MI-FNN method has the largest standard variance. This may be attributed to the inaccurate embedding dimension estimation of FNN method. Four, the proposed method performs a little higher than the TM method. This can prove that using the same parameter for all samples may not be the best parameter selection strategy. Last, the proposed method has the highest mean accuracy with the least standard variance. This is because in the proposed method, the Chebyshev distance is used to choose the embedding dimension m with the largest class separation distance. Therefore, the proposed OMPE method can extract the most discriminative MPE value as fault features. 

### 4.2. Experiment 2: Compound Bearing and Unbalance Rotor Fault

Experiment 2 is designed to prove the effectiveness of the proposed method in the application of identification compound bearing and unbalance rotor fault. The test rig (shown in Figure 13a) contains a drive motor, two rolling bearings, an unbalance rotor, a load, and a data acquisition system. The type of test bearing is MB ER-12K (the right bearing in Figure 13a). The bearing fault was simulated by replacing different fault bearings, including inner race fault (IRF), outer race fault (ORF), ball fault (BF) and combination fault (CF) (shown in Figure 13c). The unbalance fault was simulated by attaching weights on the unbalance rotor, as shown in Figure 13b. The parameters of the MFS are given in Table 3. 

In Experiment 2, ten fault types are designed, including NOR, IRF, ORF, BF, CF, unbalance fault (UN), inner race fault with unbalance fault (IRFUN), outer race fault with unbalance fault (ORFUN), ball fault with unbalance fault (BFUN), and combination fault with unbalance fault (CFUN). Each class has 100 samples and there are a total of 1000 samples (100 samples × 10 fault types). The waveforms under different fault types are shown in Figure 14. The class labels of ten fault types in Experiment 2 are shown in Table 4.

The proposed method is used in this test as described in Section 2. The whole procedure is the same as Experiment 1. The searching area of parameter k is set from 0.4 to 2.0, and the stride is set to be 0.1. The result of parameter k is shown in Figure 15. As seen the maximum value is 0.1654 and the corresponding k=0.5. Therefore, we set k=0.5. The optimum embedding dimension and optimum time delay is shown in Table 4. The mean testing accuracy of the proposed method is 80.6%. One of the classification results using the proposed method is drawn in Figure 16 (testing accuracy 162/200 = 81%).

For comparison, the FIX method and MI-FNN method are also tested. In addition, the traversal method is also used for comparison. In the TM method, we test MPE with different combinations of embedding dimension m and time delay τ. The ranges of embedding dimension *m* and τ are set as *m* = 3 to 7 and τ=1 to 10. The result of the TM method is shown in Figure 17. The highest testing accuracy is 79.15% with m=4 and τ=1. The testing accuracies of four methods are shown in Figure 18. 

In Experiment 2, three conclusions can be drawn as follows. First, MPE using FIX does not perform well with time delay τ=1 (as seen in Figure 17). This means the fixed parameter may not be suitable for all cases. Second, in Experiment 2, the MI-FNN method performs worse than the FIX method with a lower testing accuracy. This phenomenon can be attributed to the inaccurate embedding dimension estimation of the FNN method. Last, the proposed OMPE method has the highest testing accuracy with the least standard variance. Above all, our OMPE method can enhance the feature extraction ability compared with other three methods.

## 5. Conclusions

This paper proposes a parameter selection approach for MPE, namely optimized MPE (OMPE). The mutual information and improved Cao method are applied to select the time delay and embedding dimension of MPE. The performance of OMPE is validated using both simulated and experimental signals. Test results demonstrate that OMPE has the best feature extraction performance with the highest classification accuracy comparing with other three methods. The main contributions of this paper are as follows: 

(1) A parameter selection strategy is proposed to find the optimum couple of embedding dimension and time delay of MPE;

(2) OMPE can enhance the feature extraction ability with help of parameter optimization process; 

(3) Simulated and experimental signals show that OMPE method has a better performance in describing the dynamic characteristics of vibration signal.

In this preliminary study, the parameter selection strategy was validated in the MPE method. Further validation in hierarchical permutation entropy and composite multi-scale permutation entropy will be considered in our future work.

## Figures and Tables

**Figure 1 entropy-21-00170-f001:**
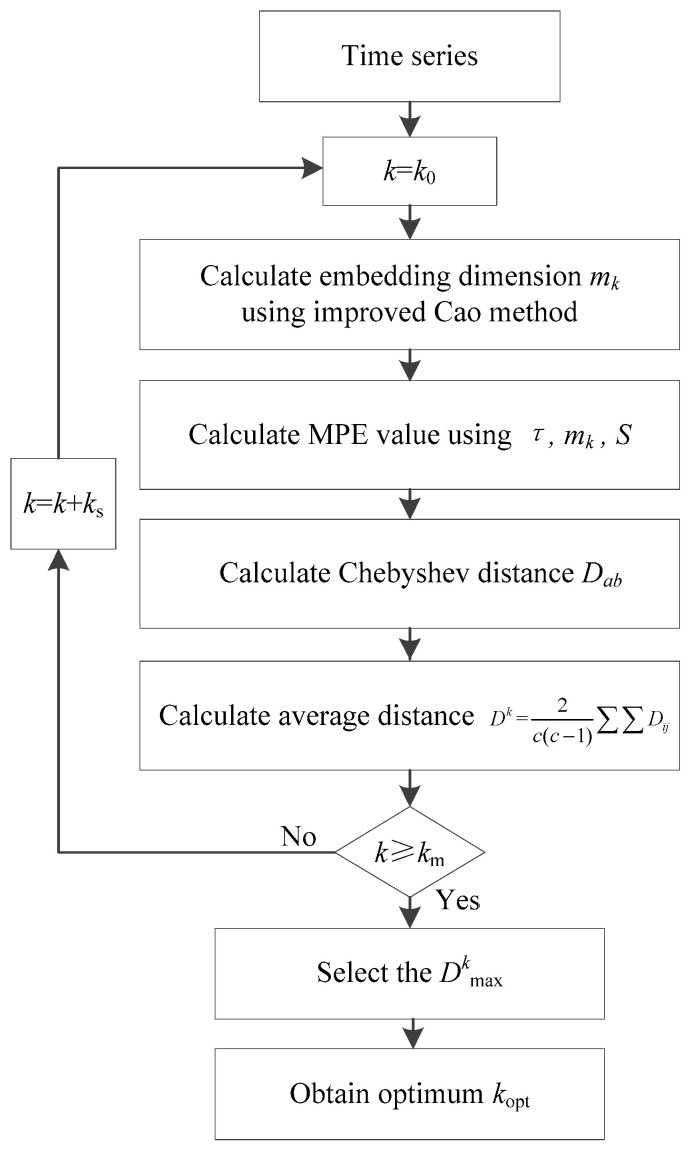
Flowchart of proposed improved Cao method.

**Figure 2 entropy-21-00170-f002:**
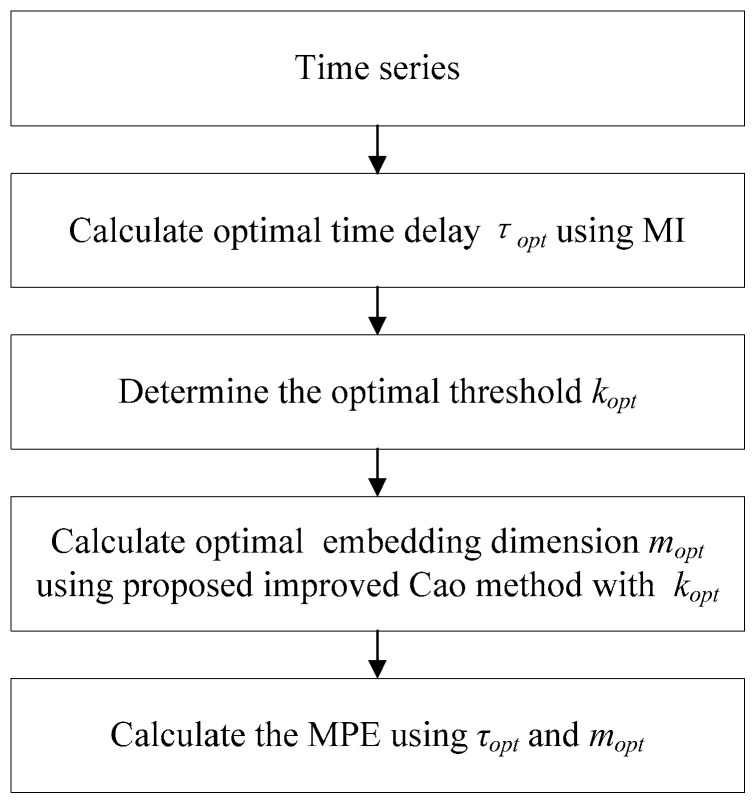
Flowchart of the proposed optimized multi-scale permutation entropy (OMPE).

**Figure 3 entropy-21-00170-f003:**
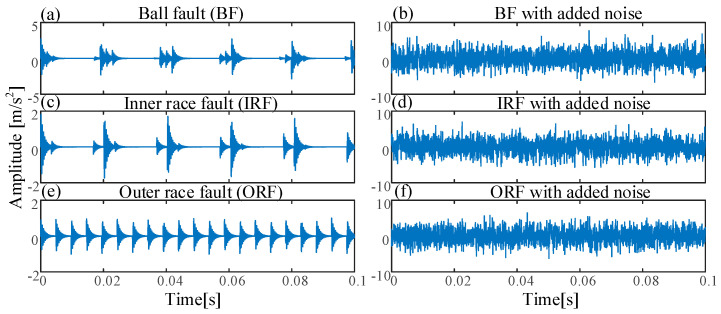
The time domains of three simulated bearing signals: (**a**) Ball fault (BF); (**b**) BF with added noise; (**c**) inner race fault (IRF); (**d**) IRF with added noise; (**e**) outer race fault (ORF); (**f**) ORF with added noise.

**Figure 4 entropy-21-00170-f004:**
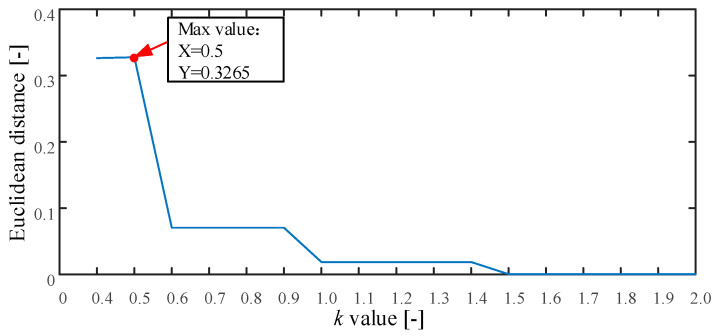
The result of parameter k for the simulated signals.

**Figure 5 entropy-21-00170-f005:**
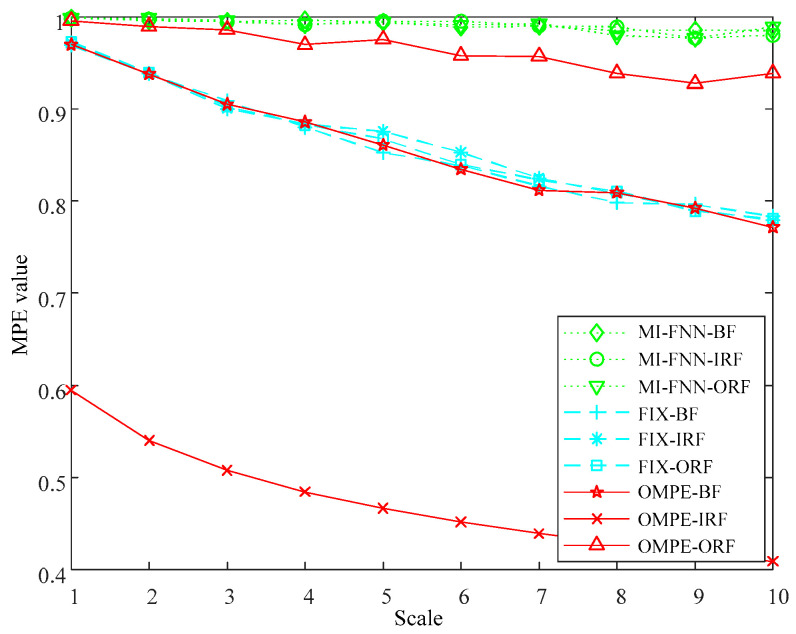
The MPE values of three parameter selection methods.

**Figure 6 entropy-21-00170-f006:**
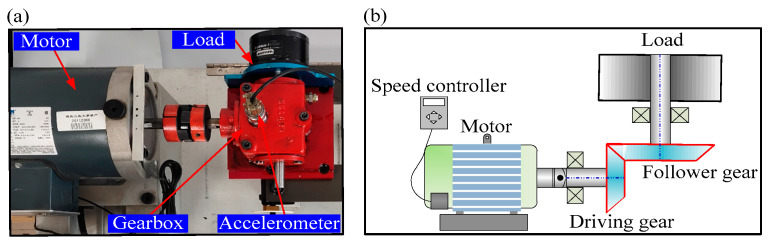
(**a**) The machinery fault simulator system; (**b**) the layout of the test rig.

**Figure 7 entropy-21-00170-f007:**
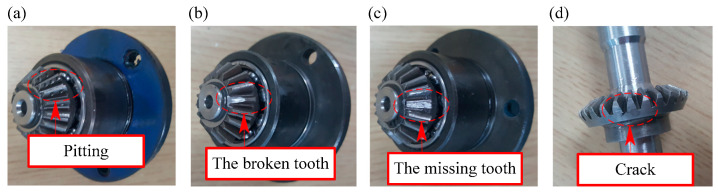
Faulty gears: (**a**) pitting in the driving tooth (PT); (**b**) broken tooth in the driving tooth (BT); (**c**) missing tooth in the driving tooth (MT); (**d**) crack in the follower tooth (CT).

**Figure 8 entropy-21-00170-f008:**
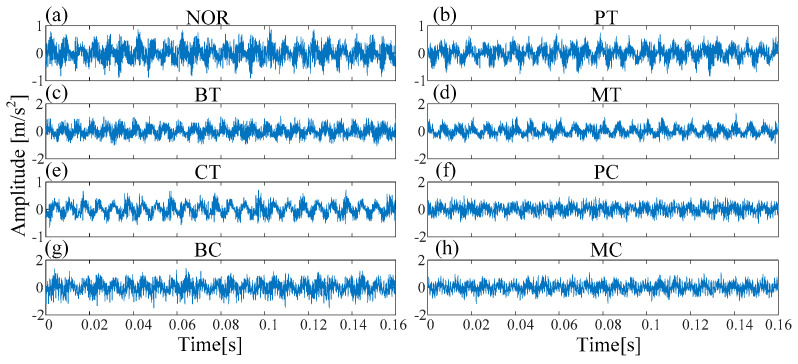
The waveforms under different fault type in Experiment 1: (**a**) Normal (**b**) pitting in the driving tooth (PT) (**b**) broken tooth in the driving tooth (BT) (**c**) missing tooth in the driving tooth (MT) (**d**) crack in the follower tooth (CT); (**f**) pitting in the driving tooth with crack in the follower tooth (PC); (**g**) broken tooth in the driving tooth with crack in the follower tooth (BC); (**h**) missing tooth in the driving tooth with crack in the follower tooth (MC).

**Figure 9 entropy-21-00170-f009:**
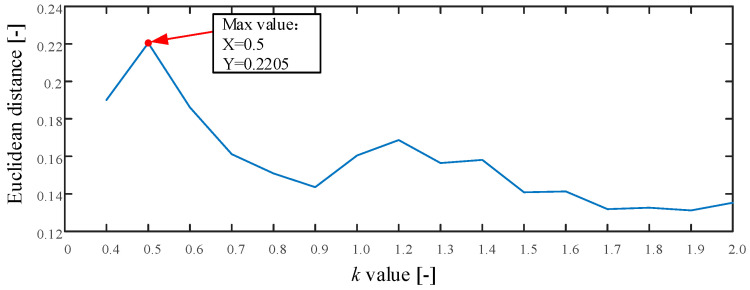
The result of parameter k in Experiment 1.

**Figure 10 entropy-21-00170-f010:**
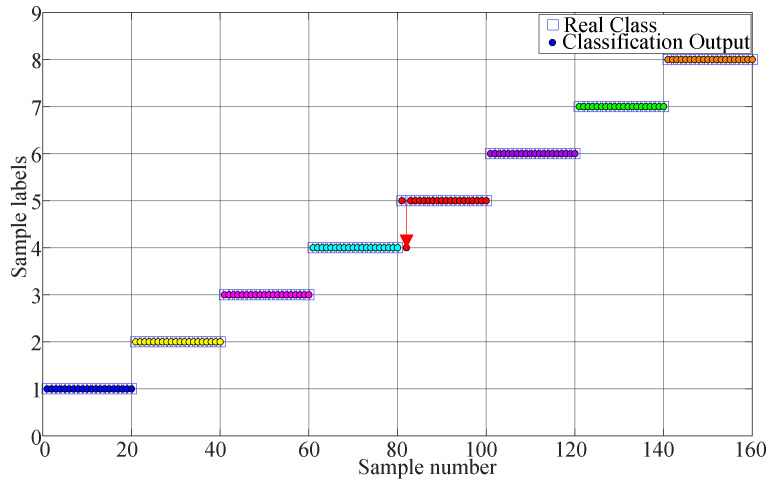
Classification result of Experiment 1 using the proposed method.

**Figure 11 entropy-21-00170-f011:**
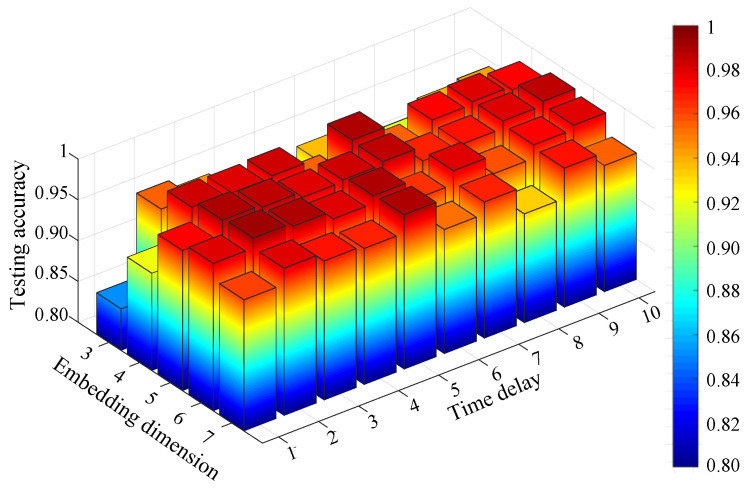
Testing accuracy of compound gear fault using the traversal method (TM) method.

**Figure 12 entropy-21-00170-f012:**
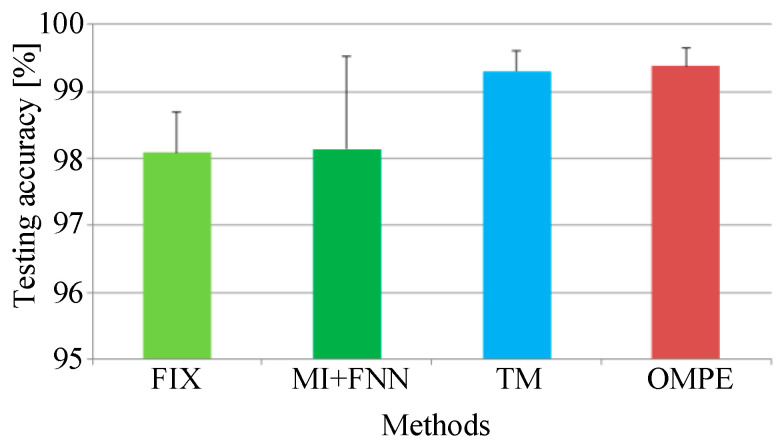
The testing accuracies of the four methods in Experiment 1.

**Figure 13 entropy-21-00170-f013:**
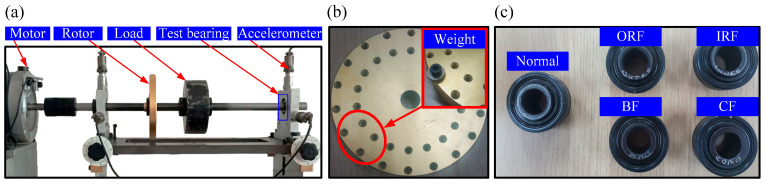
(**a**) The machinery fault simulator system; (**b**) the unbalance rotor and attaching weights; (**c**) the testing bearings with five health conditions.

**Figure 14 entropy-21-00170-f014:**
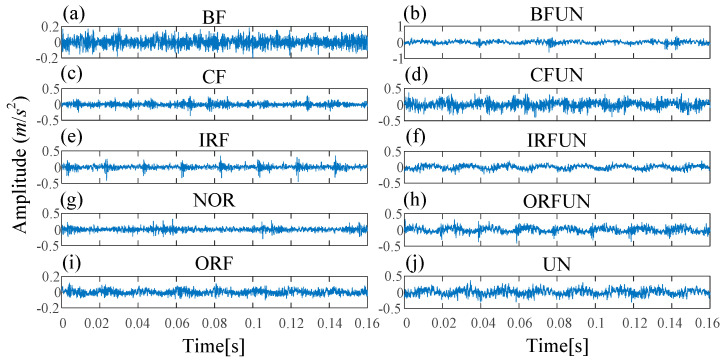
The waveforms of different fault types in Experiment 2: (**a**) BF; (**b**) ball fault with unbalance fault (BFUN); (**c**) CF; (**d**) combination fault with unbalance fault (CFUN); (**e**) IRF; (**f**) inner race fault with unbalance fault (IRFUN); (**g**) NOR; (**h**) outer race fault with unbalance fault (ORFUN); (**i**) ORF; (**j**) unbalance fault (UN).

**Figure 15 entropy-21-00170-f015:**
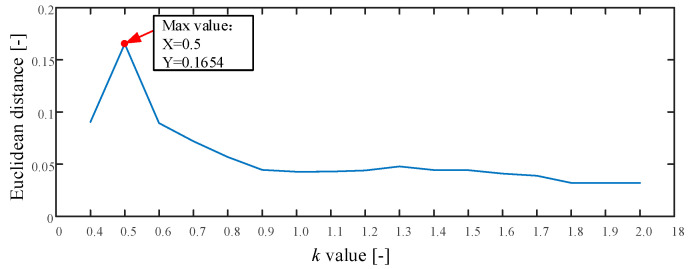
The result of parameter k in Experiment 2.

**Figure 16 entropy-21-00170-f016:**
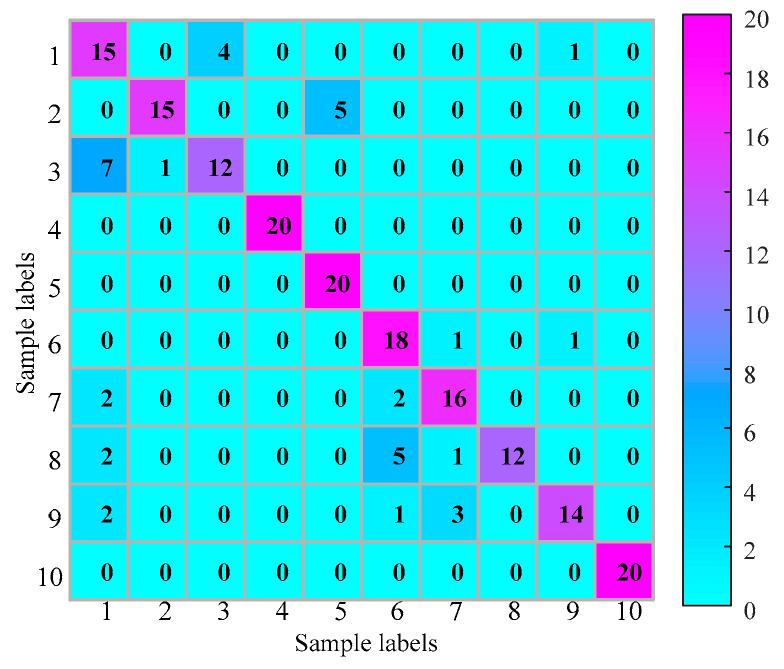
Classification result of Experiment 2 using the proposed method.

**Figure 17 entropy-21-00170-f017:**
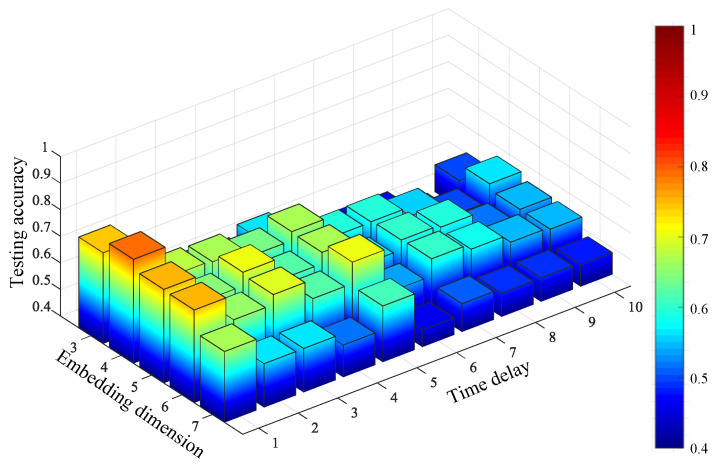
Testing accuracy of compound bearing and unbalance rotor fault using the TM method.

**Figure 18 entropy-21-00170-f018:**
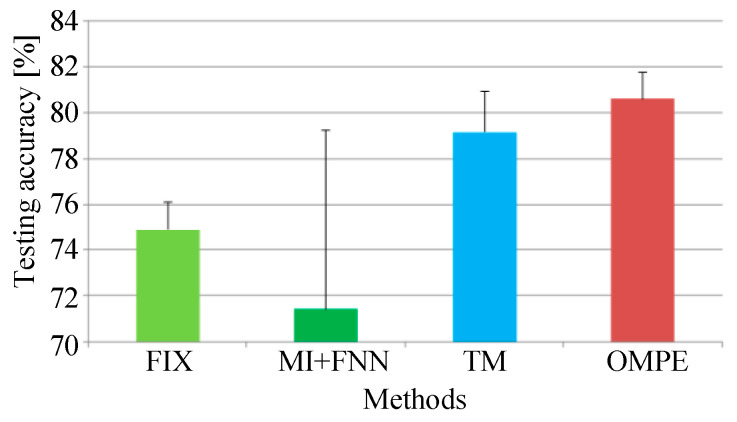
Testing accuracies of four methods in Experiment 2.

**Table 1 entropy-21-00170-t001:** The parameter settings of fixed parameter (FIX), Mutual information (MI)-false nearest neighbor (FNN) and proposed methods.

	Parameters	Embedding Dimension			Time Delay		
Methods		BF	IRF	ORF	BF	IRF	ORF
FIX method		6	6	6	1	1	1
MI-FNN method		4	4	4	3	2	2
Proposed OMPE method		6	9	5	3	2	2

**Table 2 entropy-21-00170-t002:** Optimum embedding dimension, optimum time delay and class labels in Experiment 1.

	Fault Type	PT	MT	CT	PC	MC	BC	NOR	BT
Parameters									
Embedding dimension		5	6	6	6	6	7	6	7
Time delay		3	2	3	2	2	2	2	2
Class label		1	2	3	4	5	6	7	8

**Table 3 entropy-21-00170-t003:** Working parameters of Experiment 2.

Bearing Type	MB ER-12K
Number of rolling elements	8
Rolling element diameter	0.3125 inch
Pitch diameter	1.318 inch
Contact angle	0°
Sampling frequency	12,800 Hz
Rotating speed	3000 RPM

**Table 4 entropy-21-00170-t004:** Optimum embedding dimension, optimum time delay and class labels in Experiment 2.

	Fault Type	BF	CF	IRF	NOR	ORF	BFUN	CFUN	IRFUN	ORFUN	UN
Parameters											
Embedding dimension		6	6	6	5	6	6	6	6	6	7
Time delay		3	2	3	3	2	3	3	3	3	2
Class label		1	2	3	4	5	6	7	8	9	10

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
