# Peer review of "The Optimized Multi-Scale Permutation Entropy and Its Application in Compound Fault Diagnosis of Rotating Machinery"

_entropy, 2019, doi:10.3390/e21020170_

Round 1

Reviewer 1 Report

The paper is well written and it provides an improvement of the original MPI. Therefore, I propose to publish it as it is.

Author Response

Thanks for your comments.

Reviewer 2 Report

Title: The optimized multi-scale permutation entropy and its application in compound fault diagnosis of rotating machinery

Thank you for inviting me for the review.

The  manuscript presents a method applied to fault diagnosis of rotating  machinery, which is an optimized version of the multi-scale permutation  entropy (MPE) method.

Both the method and the results showed to be of scientific interest.

Based  on the analysis of simulated and experimental data, the performance of  OMPE method was proven to be better when compared other methods used to  calculate the MPE values. Which are the FIX parameter method, MI-FNN  (Mutual Information – False Nearest Neighbor) method and Traversal  method.

The manuscript presents an appropriate literature review,  that is straight in terms of mathematical foundation.

Finally,   I understand that the manuscript could be improved if a discussion of  the algorithms were presented. Such a discussion could help both a  better understanding of the methods and also to the implementation in  the industrial environment.

Recommendations:

Line 12 correct for statistic “indicator” instead of ‘indictor’.  Also on line 27.

Line 17 when is mentioned “Cao method” its recommended to show the reference (? Maybe 24).

Line  41 The mention that: “the performance of MPE is highly depended on the  parameters setting,………”  appears excessively in the manuscript ( lines  69-71 and 281-283).

Author Response

Recommendation 1:

Line 12 correct for statistic “indicator” instead of ‘indictor’.  Also on line 27.

Response: Thanks for your comments. We have done the revisions exactly as you suggested.  The revisions are available in lines 14 and 29.

Recommendation 2:

Line 17 when is mentioned “Cao method” its recommended to show the reference? (Maybe 24).

Response: Thanks for your comments. Yes, you are right. “Cao method” is related to the Ref.[24]. However, the abstract is not allowed to add reference. We have cited Ref.[24] for Cao method in the Introduction. Meanwhile, a detailed description of Cao method [24] is given in Section 2.2.

Recommendation 3:

Line  41 The mention that: “the performance of MPE is highly depended on the  parameters setting,………”  appears excessively in the manuscript ( lines  69-71 and 281-283).

Response: Thanks for pointing this out. We have removed the inaccuracy expressions as you suggested.

The revision is available in lines 16, 43, 290.

Reviewer 3 Report

Authors show a novel approach of estimating parameters for multiscale permutation entropy in application for fault diagnosis of rolling machinery.  They introduce a method called optimized multi-scale permutation entropy (OMPE), where they implemented threshold adjusment using Chebyshev distance to find optimal parameter 'k' of Cao method of embedding dimension calculation, while delay time is estimated using mutual information.  Proposed method determined has a high potential to find its place in various applications. 

The paper is original and inspiring.  Authors put an effort on a detailed state of the art and description of used methods. Further, I highly admire that they showed analysis of the setting of their technique, further compared their technique with state of the art technique in a simulation and two experimental studies.  The paper is grammatically well-written, readable, and of minimum typos. The particular sections logically follow each other, the reader is well driven through the text.

I recommend the manuscript to be published. Here are my remarks:

-(Mandatory) Please, explain by one or two sentences  why the searching range of parameter 'k' was set from 0.4 to 2, why is not selected from 0.1 for example ? 

formal remarks

-Please, describe the abbreviations BF, IRF, ORF just close to Fig. 3. 

-All non-schematic figures should have x and y axes and axis labels that contains a quantities with corresponding units, even dimensionless, e.g. k value [-]

Author Response

Recommendation 1:

(Mandatory) Please, explain by one or two sentences why the searching range of parameter 'k' was set from 0.4 to 2, why is not selected from 0.1 for example? 

Response: Thanks for your comments. The threshold k is used to determine the embedding dimension of MPE. If k is set smaller than 0.4, the embedded dimension of MPE will be too large. This will cause that the MPE value cannot be obtained. Otherwise, if the k  is set larger than 2, a smaller embedding dimension of MPE will be generated, resulting in an inappropriate MPE values for complexity estimation. Base on above reasons, the rang of parameter k is set from 0.4 to 2.

The revision is available in page 6 (lines195 to 200).

Recommendation 2:

Please, describe the abbreviations BF, IRF, ORF just close to Fig. 3. 

Response: Thanks for your comments. We have done the revisions following your comments. The detailed revisions are available on page 6 (lines 187 and lines 206 to 208).

Recommendation 3:

All non-schematic figures should have x and y axes and axis labels that contains a quantities with corresponding units, even dimensionless, e.g. k value [-]

Response: Thanks for your comments. We have revised the figures exactly as you suggested. The revised figures include Fig.4, Fig.9, Fig.12, Fig.15 and Fig.18.
